# Estimating the effect of a scanner upgrade on measures of grey matter structure for longitudinal designs

Evelyn Medawar[1,2,3], Ronja Thieleking[1], Iryna Manuilova[1], Maria Paerisch[1], Arno Villringer[1,2,3,4,5], A. Veronica Witte[1,4,5], Frauke Beyer[1,4] *

1 Max Planck Institute for Human Cognitive and Brain Sciences, Leipzig, Germany, 2 Berlin School of Mind and Brain, Humboldt-Universität zu Berlin, Berlin, Germany, 3 Center for Stroke Research Berlin (CSB), Charité Universitätsmedizin, Berlin, Germany, 4 CRC 1052 "Obesity Mechanisms", Subproject A1, Leizpig University, Leipzig, Germany, 5 Day Clinic for Cognitive Neurology, University of Leipzig Medical Center, Leipzig University, Leipzig, Germany

* fbeyer@cbs.mpg.de

**Data Availability Statement:** The minimal underlying dataset for the region-wise analyses in this manuscript is openly available (10.17617/3.6b). Moreover, the data and the corresponding

## Abstract

Longitudinal imaging studies are crucial for advancing the understanding of brain development over the lifespan. Thus, more and more studies acquire imaging data at multiple time points or with long follow-up intervals. In these studies changes to magnetic resonance imaging (MRI) scanners often become inevitable which may decrease the reliability of the MRI assessments and introduce biases. We therefore investigated the difference between MRI scanners with subsequent versions (3 Tesla Siemens Verio vs. Skyra) on the cortical and subcortical measures of grey matter in 116 healthy, young adults using the well-established longitudinal FreeSurfer stream for T1-weighted brain images. We found excellent between-scanner reliability for cortical and subcortical measures of grey matter structure (intra-class correlation coefficient > 0.8). Yet, paired t-tests revealed statistically significant differences in at least 67% of the regions, with percent differences around 2 to 4%, depending on the outcome measure. Offline correction for gradient distortions only slightly reduced these biases. Further, T1-imaging based quality measures reflecting gray-white matter contrast systematically differed between scanners. We conclude that scanner upgrades during a longitudinal study introduce bias in measures of cortical and subcortical grey matter structure. Therefore, before upgrading a MRI scanner during an ongoing study, researchers should prepare to implement an appropriate correction method for these effects.

## Introduction

Many longitudinal neuroimaging studies of aging and development investigate changes in local grey matter volume (GMV) over time to identify biomarkers relevant to health and disease. Notably, in the past decade many large-scale studies have implemented longitudinal designs in the general population (with at least two timepoints: [1,2], second timepoint currently being acquired: [3,4].

code are available on github (https://github.com/
fBeyer89/life_upgrade). Under certain conditions,
the authors may also provide access to the whole-
brain MRI data.

**Funding:** The authors received no specific funding
for this work.

**Competing interests:** The authors have declared
that no competing interests exist.

Such longitudinal imaging studies assess within-subject differences and thereby benefit from reduction of error variance and confounding. Yet, scanner changes often become inevitable with long follow-up intervals (4–6 years) in these studies, entailing issues of reliability because of changes in signal-to-noise ratio or image intensity [5–8]. This is especially problematic in the case of two-visit longitudinal imaging studies where measurement occasion may be collinear with scanner upgrade, making it difficult to draw unbiased conclusions on within-subject change. In contrast, scanner upgrades will affect cross-sectional designs less as scanner version can be modelled like a site effect [9].

Before the follow-up of the LIFE-Adult Study, a two-visit longitudinal imaging study with a long inter-visit interval (5–7 years), we had to decide on the upgrade of the study scanner from MAGNETOM Verio to MAGNETOM Skyra [3]. At the time (end of 2017), most studies on the effects of scanner upgrades had investigated small samples (n<15) or voxel-based morphometry estimates of grey matter (GM) structure, with varying estimates of reliability and bias [10–12]. Thus, the impact of a scanner upgrade on region- and vertex-wise measures of cortical GM (thickness, area and volume) as well as subcortical GM volume still lacked quantification. Also, these studies did not take into account gradient distortion correction which has been shown to partly account for variation between scanners [13,14].

Here, we therefore investigated the difference between scanners with subsequent versions (3 Tesla Siemens Verio vs. Skyra) on the cortical and subcortical measures of GM in a large sample of healthy, young adults. Differences between the systems included the changes introduced by software and hardware upgrades (update to syngo MR E11 software, a new Tim 4G body coil and installation of DirectRF) and side-specific variations in the scanner hardware.

Using the validated longitudinal *FreeSurfer* stream, we expected the reliability of whole-brain and regional GM measures to be similar to previous studies investigating between-site reliability [15–17]. Based on previous upgrade studies, we hypothesized a systematic bias with varying effect sizes and direction in cortical and subcortical regions [10,18]. Finally, we expected gradient distortion correction to improve reliability and reduce bias.

## Methods

### Sample

121 healthy participants (median age = 28 years, range = 19–54 years; 61 females) were scanned on two different 3 Tesla MRI scanners MAGNETOM Verio syngo MR B17A (Siemens Healthcare, Erlangen, Germany) and MAGNETOM Skyra fit syngo MR E11C (Siemens Healthcare, Erlangen, Germany). Scanners are referred to as Verio and Skyra throughout the manuscript. Due to a pending version upgrade of the Verio scanner, all participants were first scanned at the Verio and then at the Skyra scanner. The median time between sessions was 7.7 weeks (range: 0.5–18.2 weeks).

5 participants did only participate in the first scanning session at the Verio and were therefore excluded in the following analysis. The study was approved by the local ethics committee at the University of Leipzig and all participants gave written informed consent according to the Declaration of Helsinki.

### Differences between acquisitions

The upgrade from Verio to Skyra includes an extensive retrofit of hardware and software components (e.g. new body coil, new RF transmit and receive signal transmission system and change to syngo MR E11 software). For more information on the upgrade, see the Siemens product brochure saved in *https://github.com/fBeyer89/life_upgrade*. In this study, the Verio

did not undergo an actual upgrade and the scanners therefore differed in the main B0-field and other hardware components. See Table 1 for a summary of differences between scanners.

On both scanners, anatomical T1-weighted imaging was performed with the vendor-implementation of the magnetization-prepared rapid gradient-echo (MPRAGE) sequence (TR = 2300 ms,TE = 2.98 ms, TI = 900 ms, parallel imaging: GRAPPA with factor 2 and adaptive coil combination, flip angle: 9˚, imaging matrix 256 x 240 x 176 and voxel size = 1 $mm^3$, with prescan normalize option) according to the ADNI-3 protocol [19]. On both scanners, a 32 channel head coil was used. On the Skyra scanner, both online 3D gradient distortion-corrected images (D) and images not corrected for distortions (ND) were available. The Verio scanner delivered the images without gradient-distortion correction (ND).

## Image processing

**FreeSurfer analysis.** To extract reliable volume and thickness estimates, we processed the T1-weighted images with the longitudinal stream in *FreeSurfer* [15]. Within this pipeline, an unbiased within-subject template space is created using robust and inverse consistent registration [20,21]. The longitudinal stream increases the reliability of cortical and subcortical GM estimates compared to the cross-sectional stream and is thus appropriate for longitudinal studies [17]. We used *FreeSurfer* version 6.0.0p1 with the default parameters *recon-all -all -parallel -no-isrunning -openmp 8*, which include non-parametric non-uniform intensity normalization with the MINC tool *nu_correct*. We ran the *recon-all* longitudinal stream with the Verio ND and Skyra ND images, and additionally with Verio ND and Skyra D images.

**Gradient distortion correction.** Gradient distortion correction has been shown to contribute to measurement error in repeated sessions of anatomical brain imaging [6]. Accordingly, correcting for distortion correction can improve the reproducibility of intensity data significantly [13]. For the Verio scanner, the vendor provided no online distortion correction while the Skyra system offered online 3D-distortion correction. To assess the effect of this processing step on reliability and bias, we applied an identical tool for offline gradient distortion correction on the ND sequences from both scanners.

Gradient unwrapping calculates the geometric displacement based on the spherical expansion of the magnetic gradient fields and applies it to the image [13,22]. We used the *gradunwarp* implementation [(*https://github.com/Washington-University/gradunwarp*)] v1.1.0 in Python 2.7. We visually compared the original and the *gradunwarp* result files to determine

**Table 1. Information on hardware, software and acquisition parameters used on the Verio and Skyra scanner.**

| MRI scanner | Verio | Skyra |
|---|---|---|
| Nominal field strength | 3 Tesla | 3 Tesla |
| System software version | syngo MR B17A | syngo MR E11C |
| Tx/Rx coil | Body coil: Tim Tx Trueform [102x32] 32-channel head coil | Body coil:Tim 4G [204x64] 32-channel head coil |
| Transmission amplifier reference amplitude (mean/standard deviation in V) | 432/27.8 | 311/15.7 |
| Gradient system | VQ Gradients 45 mT/m @ 200 T/m/s | VQ Gradients 45 mT/m @ 200 T/m/s |
| Coil combination algorithm | adaptive coil combination | adaptive coil combination |
| Imaging sequence | Vendor-implementation of MPRAGE with standardized parameters according to ADNI-3 | Vendor-implementation of MPRAGE with standardized parameters according to ADNI-3 |
| Parallel imaging method | Grappa with iPAT: 2 | Grappa with iPAT: 2 |
| TE (ms) | 2.98 | 2.98 |
| Acquisition time (min:sec) | 5:10 | 5:10 |

the appropriate number of sampling points and interpolation order. Based on this, we chose 200 sampling points and 4th order interpolation (—fovmin -0.2—fovmax 0.2—numpoints 200 —interp_order 4) because this yielded most similar intensity distributions. After unwarping, we repeated *FreeSurfer*'s cross-sectional and longitudinal stream for these images. Then, we assessed the reliability and bias in cortical and subcortical ROI measures between the *gradunwarp* distortion corrected Skyra ND and Verio ND images.

**Outcomes.** We selected cortical thickness (CT), area (CA) and volume (CV) estimates for regions of interests defined by the Desikan-Killiany (DK) cortical parcellation (64 ROI for both hemispheres) as outcomes. Subcortical volumes were extracted from *FreeSurfer*'s subcortical segmentation ("aseg.mgz", 18 bilateral ROI). We analyzed all ROI per hemisphere. Subcortical volumes were not adjusted for head size because during the longitudinal stream, both images are normalized to the same head size.

**Quality assessment.** We visually checked the cross-sectional as well as the longitudinal runs for errors in white matter segmentation and misplaced pials [23]. There were 17 cases where the pial surface expanded into non-brain tissue. These were corrected by either editing the brainmask in the longitudinal template or by correcting the cross-sectional runs. After correction, we re-ran the longitudinal template creation step and the longitudinal timepoints. No issues regarding white matter segmentation were noticed.

To quantify potential differences in image quality between scanners, we compared different quality control measures provided by *mriqc* (version 0.15.0) [24]. We used signal-to-noise ratio (SNR) to assess overall signal quality and compared contrast-to-noise ratio (CNR) to quantify the difference between grey and white matter intensity distributions. Furthermore, we used coefficient of joint variation (CJV) which also reflects grey-to-white matter contrasts and entropy focus criterion (EFC) to describe the amount of ghosting and blurring induced by head motion [24]. We performed *mriqc* on the Verio ND, Skyra ND and Skyra D images.

## Analysis

In the main analysis, we compared Verio ND and Skyra ND as they correspond to the same stage of image reconstruction and are therefore most comparable (red arrow in Fig 1). Additionally, we investigated the effects of offline *gradunwarp* distortion correction on Verio ND and Skyra ND images (blue arrow in Fig 1). In a supplementary analysis, we compared Verio ND and Skyra D outcomes as these are the default reconstructions available at the respective scanners (green arrow in Fig 1).

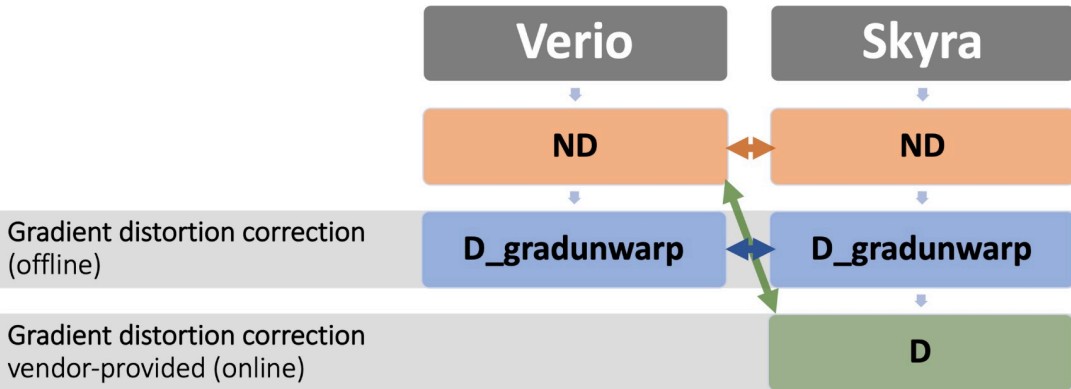

**Fig 1. Overview of the acquisitions and the performed analyses.** Orange: Main analysis comparing Verio ND and Skyra ND. Blue: Analysis of gradunwap Verio ND and Skyra ND. Green: Additional analysis of default scanner outputs Verio ND and Skyra D.

All statistical analysis were performed in R version 3.6.1 [25]. The package *fsbrain v.0.3.0* was used to plot vertex- and ROI-wise results [26].

**Reliability and percent difference of cortical and subcortical GM measures.** To assess the reliability of the grey matter (GM) estimates, we calculated the intra-class correlation coefficient (ICC), an established measure of agreement between raters. The ICC is calculated as the proportion of overall variance that is explained by between-subject variance, and thereby gives an estimate of the variance introduced by systematic differences and error between raters [27].

$$\rho_{3A} = \frac{\sigma_{r^2}}{\sigma_{r^2} + \Theta_{c^2} + \sigma_{v^2}}$$

Here, $\sigma_{r^2}$ is the population variance, $\Theta_{c^2}$ is the variance of fixed biases and $\sigma_{v^2}$ is the error variance. We used the two-way mixed effect ICC model for single measures with absolute agreement [28], implemented in the package *psy* to calculate ICC for each cortical DK and subcortical ROI and reported the estimate and 95% confidence interval, derived by bootstrapping. According to [29], we considered an ICC below .4 to be poor, between .40 and .59 to be fair; .60 and .74 to be good and between .75 and 1.00 to be excellent.

ICC depends on the between-subject variance (i.e. when between-subject variance is low, ICC decreases even if rater bias remains similar) and does not provide an estimation of bias and difference between measurements. Therefore, we used Bland-Altman plots with 95% limits of agreement to visually compare the agreement between the two scanners [30].

To quantify the relative difference of GM measures between scanners, we calculated percent difference (PD) (also termed variability error [17,31]). We calculated the mean of the PD for each ROI $j$ across $n$ participants according to

$$PD_j = \frac{2}{n} \sum_{i=1}^{n} \frac{V_{ij} - S_{ij}}{V_{ij} + S_{ij}}$$

where $V_{ij}$ is the GM measure of a ROI measured on the Verio, $S_{ij}$ is the GM measure of a ROI measured on the Skyra.

Finally, we performed paired t-tests to inform about the direction and statistical significance of potential systematic differences between scanners. Here, we used Benjamini-Hochberg correction to adjust p-values per cortical GM measure and deemed differences to be significant at $p_{adj} < 0.05$ [32]. We reported T-value, uncorrected and corrected p-values.

We assessed the improvement by comparing the ICC and PD measures of CT and subcortical volume between the gradient distortion correction analysis (*gradunwarp* Skyra ND Verio ND), the original analysis (Skyra ND, Verio ND) and the secondary analysis (Skyra D, Verio ND) with paired t-tests or ANOVA. A p < 0.05 was interpreted as significant.

**Vertex-wise estimation of reliability and percent difference.** For whole-brain visualization, we performed vertex-wise calculations on the fsaverage template following [33] in Matlab version 9.7 (2019b). We calculated ICC and PD for cortical thickness, area and volume to visualize reliability and difference between scanners on a vertex-wise level.

**Quality metrics.** For the quality metrics from *mriqc*, we used linear mixed models (LMM) to assess differences between scanners (Verio, Skyra) and acquisitions (D, ND) using *lmerTest*. Significance was defined based on model comparisons (using Chi-square test with R's anova) between LMM including either scanner or acquisition as a fixed effect and null models only including the random effects of subject. Significance was defined as p < 0.05. We also tested whether CNR was associated with regional CT, independent of scanner, using a LMM with both factors. We reported $\beta$ estimates, raw and Benjamini-Hochberg adjusted p-values.

## Results

### Differences in cortical GM measures between scanners

Figs 2–4 summarize the results for CT, CA and CV, respectively.

Overall, the ICC or scan-rescan reliability was excellent (CT: mean = 0.91, min = 0.82, max = 0.97; CA: mean = 0.98, min = 0.91, max = 1; CV: mean = 0.98, min = 0.93, max = 0.99).

The PD was around 2–3% for all measures (CT: mean = -0.2, min = -2.06, max = 1.79; CA: mean = -0.82, min = -3.42, max = 2.99; CV: mean = -1.51, min = -3.3, max = 2.8) with a significant bias. The most pronounced bias was found for CT, with lower CT in Verio compared to Skyra in medial frontal and central regions, and higher values in Verio compared to Skyra in lateral occipital and inferior temporal regions. For CA and CV, the bias pattern was more related to gyrification, with higher CA/CV for Skyra compared to Verio in sulci, and the reverse pattern in gyri (see Fig 5). Overall, the CT bias direction seems to follow a frontal-to-lateral-occipital pattern, and CA and CV differed between gyral-sulcal areas.

Bland-Altman plots confirmed the bias of Verio versus Skyra measurements. Exemplary plots of superior frontal and lateral occipital regions are shown in Fig 6.

For the superior frontal region, CT, CA and CV are larger for Skyra compared to Verio with 95% of CT differences were between -0.04 and -0.12 mm, 95% of CA differences were between -103.12 and -285.25 $mm^2$, and 95% of CV differences were between -694.23 and -1841.61 $mm^3$. The inverse pattern was present in the lateral occipital region for CT with 95% of CT differences were between 0.03 and -0.07 $mm$. Here, 95% of CA differences were between -55.32 and -156.55 $mm^2$, and 95% of CV differences were between -86.04 and -772.28 $mm^3$. Accordingly, paired t-tests indicated systematic differences between scanners for the majority of regions of interest (FDR-corrected, CT: 67.2% of all 64 bilateral ROI, CA: 92.2%, CV: 90.6%).

For detailed results per cortical region see Tables 1–3 in S1 File and for vertex-wise ICC maps see Figs 2, 4 and 6 in the S1 File.

### Differences in subcortical measures between scanners

As shown in Table 2, subcortical regions, similar to cortical areas, showed excellent reliability for all regions of interest (mean = 0.95, min = 0.81, max = 0.99). The PD was around 2–3% (mean = 2.76%, min = 1.34%, max = 9.56%), with an outlying PD of 9.5% for left Accumbens.

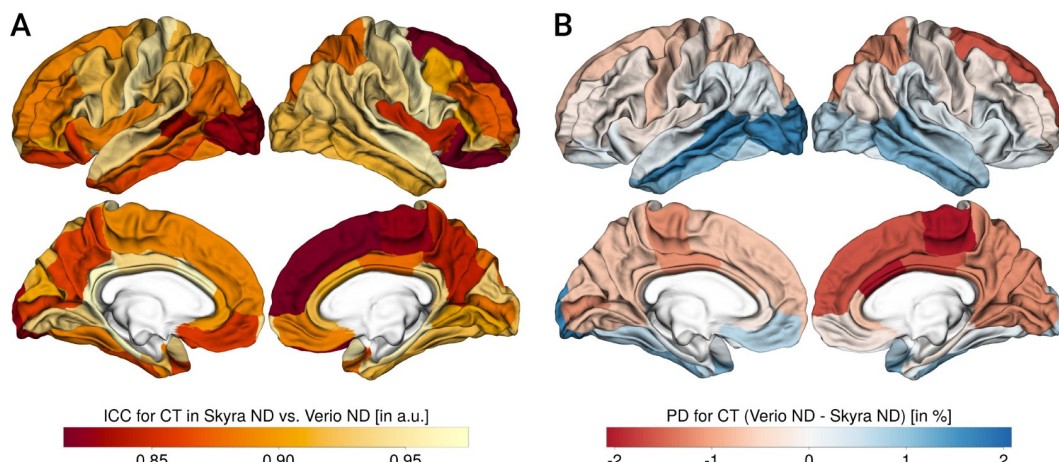

**Fig 2. Reliability and percent differences for cortical thickness (CT).** A: CT ICC, B: CT PD (for each panel, left column shows lateral and medial view of left hemisphere, right column shows lateral and medial view of right hemisphere), negative values:Skyra>Verio, positive values: Verio>Skyra.

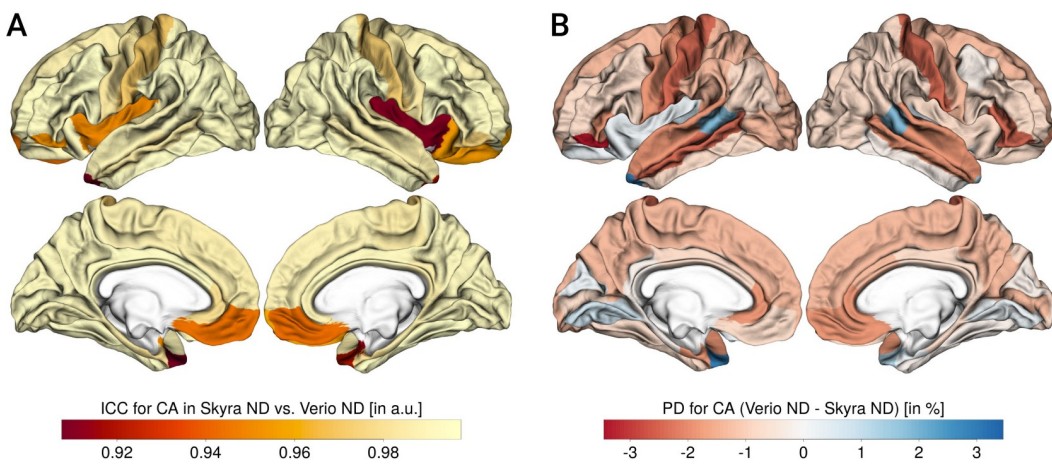

**Fig 3. Reliability and percent differences for cortical area (CA).** A: CA ICC, B: CA PD (for each panel, left column shows lateral and medial view of left hemisphere, right column shows lateral and medial view of right hemisphere), negative values: Skyra>Verio, positive values: Verio>Skyra.

Higher values were measured on Skyra compared to Verio for all regions, and these were significant for most regions (FDR-corrected, 85.7% of all 14 bilateral ROI).

Bland-Altman plots for subcortical regions confirmed the systematic bias and further indicated that differences in variability between subjects influenced ICC estimates. For example, there was high between-subject variability in the Thalamus, so that, despite large differences between measurements, ICC was high. Similarly, differences between scanners were less pronounced in the Accumbens, yet, due to lower between-subject variability the ICC of this region was lower (see Fig 7 and Fig 1 in the S1 File). For the Accumbens, 95% of differences between scanners were between -143.02 and 37.38 $mm^3$, for Thalamus 95% of differences were between -914.4 and 233.85 $mm^3$.

## QA measures

First, we compared SNR, CNR, CJV and EFC, four quality measures from *mriqc* between Verio ND and Skyra ND acquisitions. We aimed to determine whether differences in basic signal properties might underlie the observed differences in measures of GM structure.

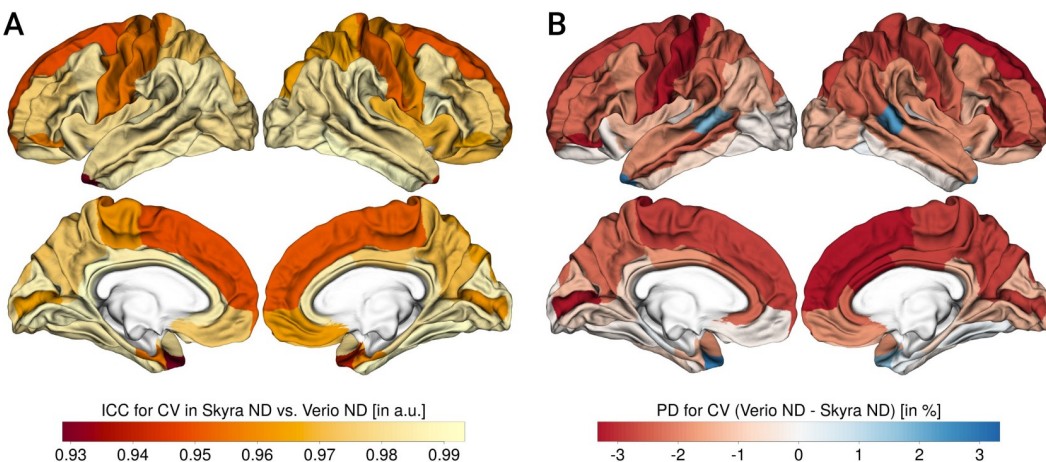

**Fig 4. Reliability and percent differences for cortical volume (CV).** A: CV ICC, B: CV PD (for each panel, left column shows lateral and medial view of left hemisphere, right column shows lateral and medial view of right hemisphere), negative values:Skyra>Verio, positive values: Verio>Skyra.

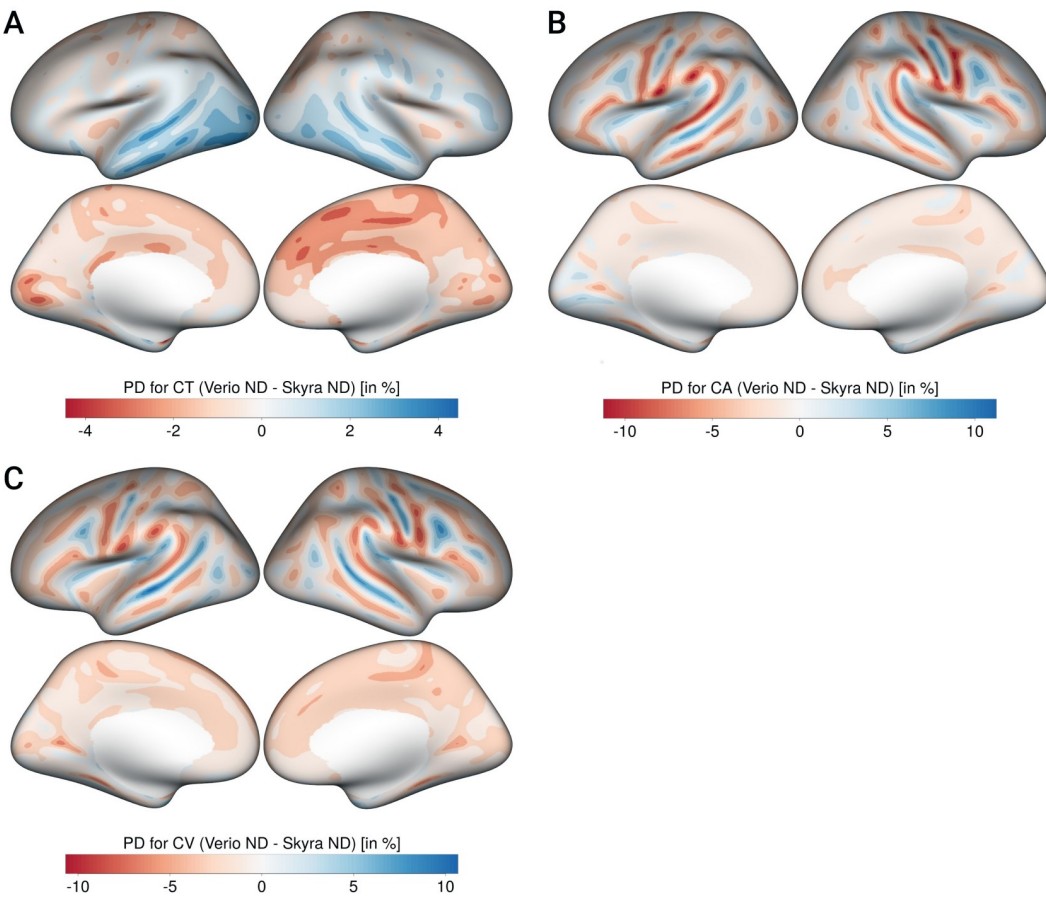

**Fig 5.** Bias patterns for CT (A), CA (B) and CV (C) shown on inflated white surface. Left column in each panel shows lateral and medial view of left hemisphere, right column shows lateral and medial view of right hemisphere), negative values: Skyra>Verio, positive values: Verio>Skyra. CT: Cortical thickness, CA: Cortical area, CV: Cortical volume.

We found that there was no significant difference in SNR between Verio and Skyra ($\beta$ = 0.06, p = 0.07). Yet, Verio ND T1-weighted images had higher CNR ($\beta$ = 0.2, p < 0.001), lower EFC ($\beta$ = -0.03, p < 0.001) and lower CJV ($\beta$ = -0.02, p < 0.001) compared to Skyra ND images, also see Fig 8. This indicates higher contrast between WM and GM and less blurring on the Verio scanner.

Similar to [11], we investigated whether increased CNR would predict differences in CT. Here, we found that higher CNR across both scanners was associated with higher CT for most regions (see Fig 9, left panel). Moreover, scanner predicted CT independent of CNR in the same regions as shown above (see Fig 9, right panel, and Table 4 in the S1 File).

Fig 10 shows the association of CNR and CT for two exemplary regions with contrary scanner effects (superior frontal and lateral occipital).

## Effect of offline gradient distortion correction

We examined whether the differences in cortical and subcortical GM measures arise from the difference in gradient distortion between the two scanners. We corrected both ND files using vendor-provided information on gradient distortions using gradunwarp.

Fig 11 shows the results for CT derived from the gradunwarp distortion corrected data (also see Table 5 in the S1 File). The ICC was excellent throughout all ROI (mean = 0.91,

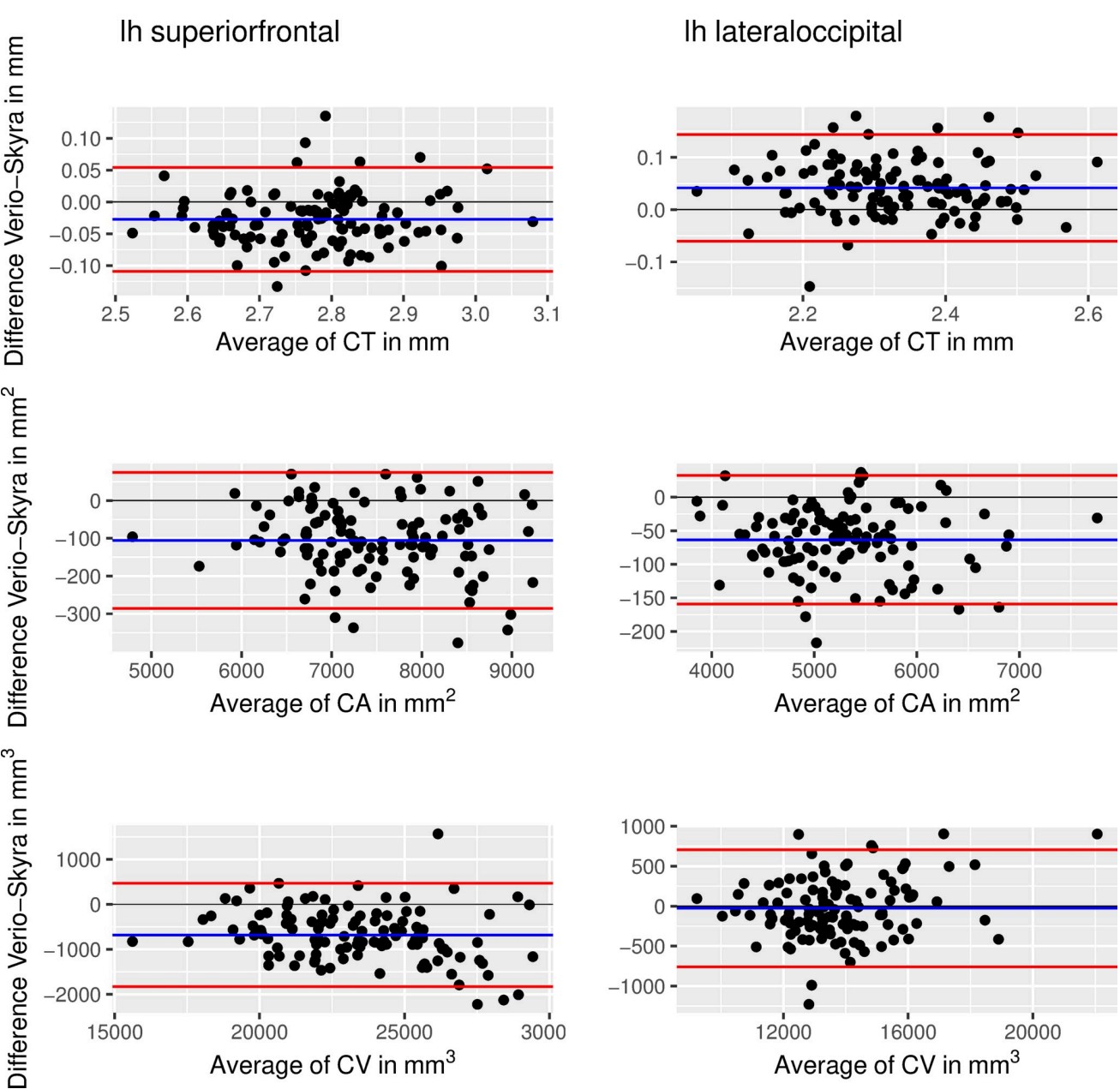

**Fig 6.** Bland-Altman plot showing differences of Verio ND–Skyra ND against means for superior frontal (left column) and lateral occipital cortex (right column) of the left hemisphere. Top row shows cortical thickness, middle row shows cortical area and bottom row shows cortical volume. Limits of agreement at 95% of standard deviation).

min = 0.8, max = 0.98), and as expected, it was higher for the gradient distortion corrected data compared to the analysis of Verio ND vs Skyra ND (mean ICC *gradunwarp* Skyra ND vs Verio ND: 0.913, mean ICC Skyra ND vs Verio ND: 0.906, paired t-test: T = -3.22, p 0.002).

Gradient distortion correction reduced PD to 1–2% (mean = -0.26%, min = -1.84%, max = 1.9), which was significant compared to ND data (mean PD *gradunwarp* Skyra ND vs Verio ND: 0.64, mean PD Skyra ND vs Verio ND: 0.72, paired t-test: T = 2.38, p = 0.02). Yet, the inferior-superior pattern of biases remained similar, and there were still significant differences after *gradunwarp* for the majority of regions of interest (FDR-corrected, 62.5% of 64

**Table 2. Reliability (mean ICC, lower and upper ICC 95% confidence interval) and percent difference (PD) for subcortical grey matter volumes, separated by hemisphere (T<0 reflects Skyra>Verio, T>0 reflects Verio>Skyra).**

| Region of interest | hemi | ICC | lower ICC | upper ICC | PD | T | p | FDR-corrected p |
|---|---|---|---|---|---|---|---|---|
| Thalamus | Left | 0.97 | 0.97 | 0.98 | 1.77 | -11.38 | 0.00 | **0.00** |
| Thalamus | Right | 0.98 | 0.97 | 0.98 | 1.51 | -10.06 | 0.00 | **0.00** |
| Caudate | Left | 0.99 | 0.99 | 0.99 | 1.43 | -9.34 | 0.00 | **0.00** |
| Caudate | Right | 0.98 | 0.98 | 0.98 | 2.00 | -14.23 | 0.00 | **0.00** |
| Putamen | Left | 0.98 | 0.97 | 0.98 | 1.78 | -8.69 | 0.00 | **0.00** |
| Putamen | Right | 0.99 | 0.98 | 0.99 | 1.34 | -6.64 | 0.00 | **0.00** |
| Pallidum | Left | 0.96 | 0.96 | 0.97 | 2.51 | -3.48 | 0.00 | **0.00** |
| Pallidum | Right | 0.95 | 0.94 | 0.96 | 2.79 | -3.34 | 0.00 | **0.00** |
| Hippocampus | Left | 0.96 | 0.95 | 0.96 | 2.20 | -13.54 | 0.00 | **0.00** |
| Hippocampus | Right | 0.97 | 0.97 | 0.98 | 1.71 | -8.03 | 0.00 | **0.00** |
| Amygdala | Left | 0.94 | 0.92 | 0.96 | 3.19 | -0.56 | 0.57 | 0.57 |
| Amygdala | Right | 0.94 | 0.93 | 0.96 | 2.90 | -1.55 | 0.12 | 0.13 |
| Accumbens | Left | 0.81 | 0.79 | 0.86 | 9.56 | -10.25 | 0.00 | **0.00** |
| Accumbens | Right | 0.95 | 0.93 | 0.95 | 3.96 | -4.85 | 0.00 | **0.00** |

P-values are shown uncorrected, and FDR-corrected where **bold** indicates p<0.05.

bilateral cortical ROI). In addition, we saw that the systematic bias was largest when comparing the default scanner output images Skyra D (online gradient distortion correction) and Verio ND (mean ICC *gradunwarp* Skyra ND vs Verio ND: 0.91, mean ICC Skyra ND vs Verio ND: 0.91, mean ICC Skyra D vs Verio ND: 0.89, ANOVA F-test: F = 3.7, p = 0.026).

Table 3 shows the results for subcortical volumes derived from the gradient distortion corrected data.

The ICC is excellent in all regions, similar to the cortical analysis (mean = 0.95, min = 0.81, max = 0.99). For subcortical volumes, gradient distortion correction did not lead to a further improvement in ICC (mean ICC *gradunwarp* Skyra ND vs Verio ND = 0.95, mean ICC Skyra ND vs Verio ND = 0.95, paired t-test: T = 1, p = 0.34).

The PD was around 2–3% (mean = 2.81%, min = 1.39%, max = 9.23%) and did not differ from the original analysis (mean PD *gradunwarp* Skyra ND vs Verio ND = 2.81%, mean PD Skyra ND vs Verio ND = 2.76%, paired t-test: T = -1.06, p = 0.31). There were significant differences after *gradunwarp* for all regions of interest (FDR-corrected, 100% of 14 bilateral subcortical regions).

## Discussion

### Summary

In this paper, we aimed to investigate the reliability and bias in GM structure induced by a scanner upgrade in a longitudinal study. We compared outcomes of *FreeSurfer*'s longitudinal pipeline between two different MRI scanners with subsequent versions. We found between-scanner reliability measured with ICC to be excellent. Yet, Bland-Altman plots and paired t-tests revealed statistically significant differences, i.e. biases, in CT and subcortical GM volumes, as well as in CA and CV in a large number of regions. Offline correction for gradient distortions based on vendor-provided gradient information reduced this bias significantly, yet it was not fully removed. T1-imaging based quality measures differed systematically between scanners. We conclude that scanner upgrades during a longitudinal study introduce bias in measures of cortical and subcortical grey matter structure and make it difficult to detect true effects when these are subtle like in the case of healthy aging, e.g. ~ 1% annual hippocampal

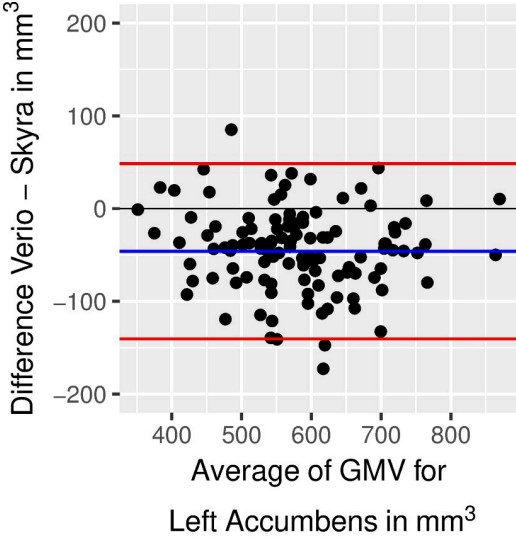

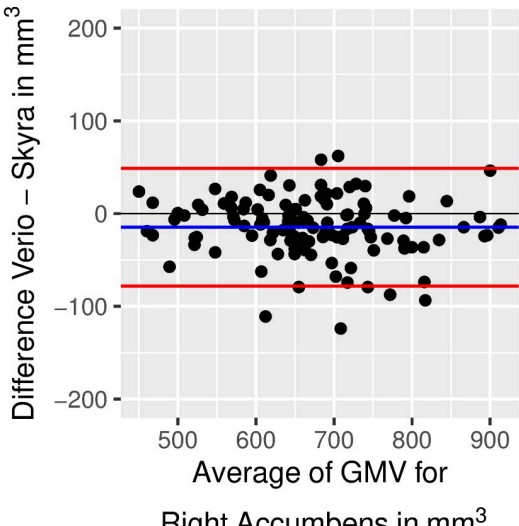

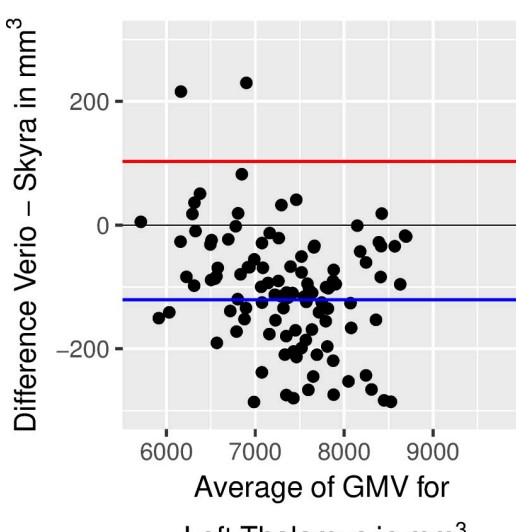

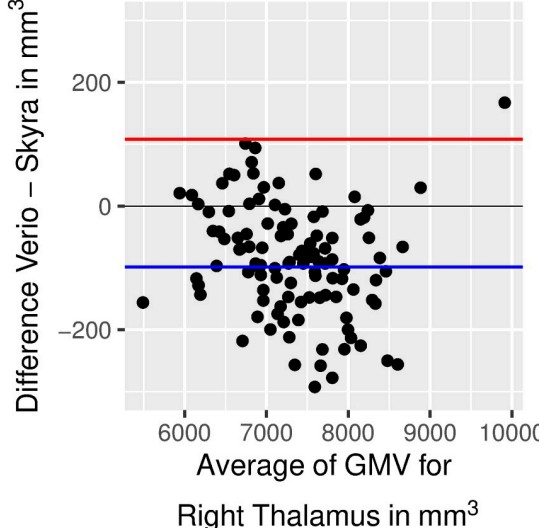

**Fig 7.** Bland-Altman plot showing differences of Verio ND–Skyra ND against means for left Accumbens (top row) and left Thalamus (bottom row). Limits of agreement at 95% of standard deviation).

volume loss in older healthy adults [34]. Therefore, before upgrading a MRI system during an ongoing longitudinal study, researchers should prepare to implement an appropriate correction method, such as deriving scaling factors from repeated measures before/after the upgrade or statistical adjustment methods.

## Comparison to previous reliability studies

The results of our study are in line with previous findings which have indicated systematic effects of scanner upgrade on GM imaging outcomes [10,18,35,36].

In recent studies, scanner upgrades induced a significant bias in cortical and subcortical GM measures, [37,38], while in one study no systematic bias was reported for hippocampus

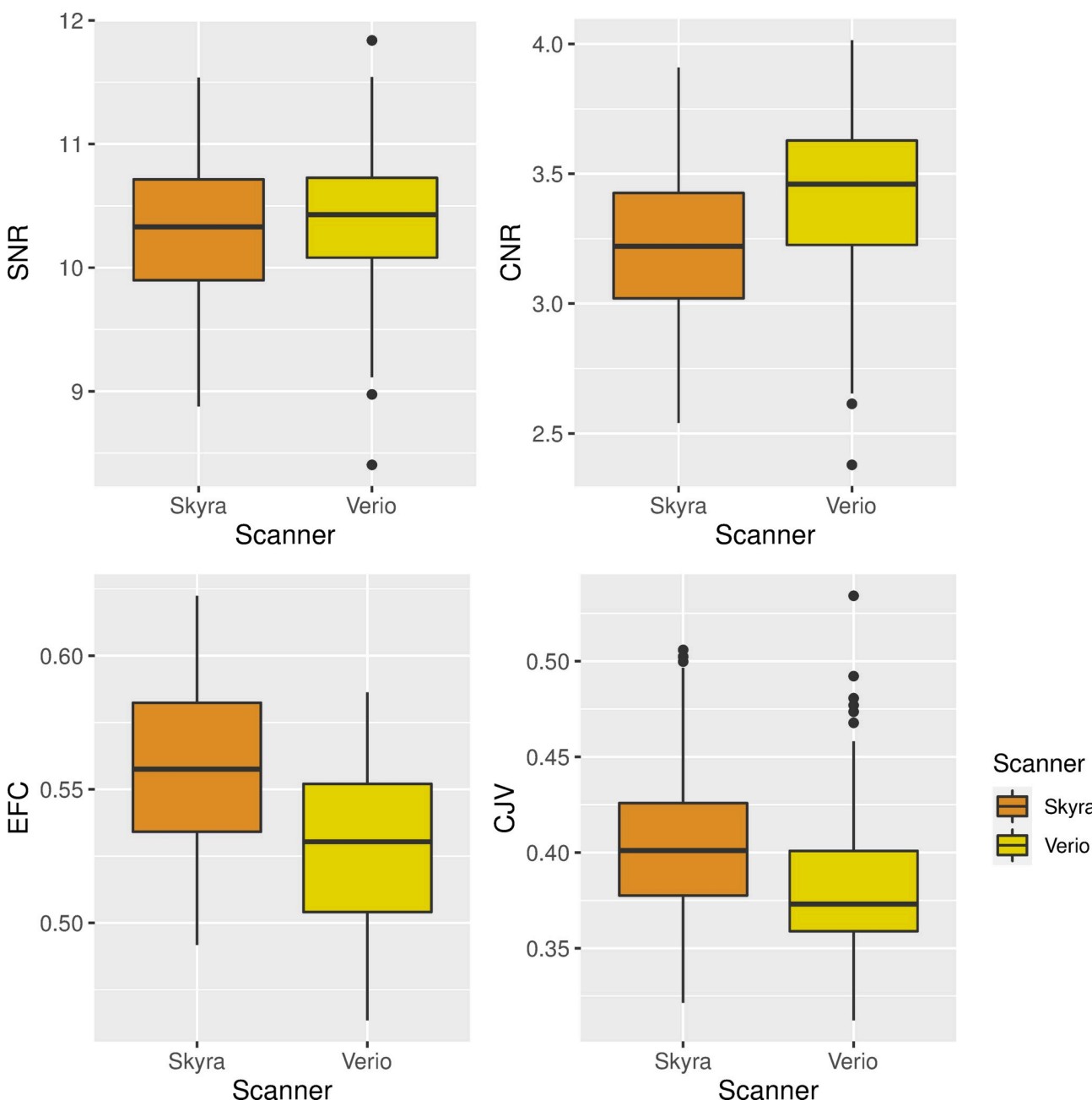

**Fig 8.** Quality metrics (SNR and CNR (left/right in upper row), EFC and CJV (left/right in bottom row)) compared between Skyra ND (orange) and Verio ND (yellow) acquisitions, showing overall higher data quality on the Verio scanner.

volume [39]. Similar to our findings, ICC values for cortical measures were good to excellent. While the size of biases was comparable to our results (around 1–6% for cortical PD for CV and CT in [37]), the location of the biased regions was different. In [37,38] GM estimates in prefrontal and temporal regions increased with the upgrade, which might be driven by upgrade-related increases in SNR in these regions, which typically show relatively poor within-subject reliability [31,33]. In our study, we found a medial-frontal to lateral-occipital gradient, with medial-frontal CT as well as subcortical volumes biased towards higher CT and GM volume in Skyra compared to Verio, while lateral-occipital CT was higher in Verio. For CA and

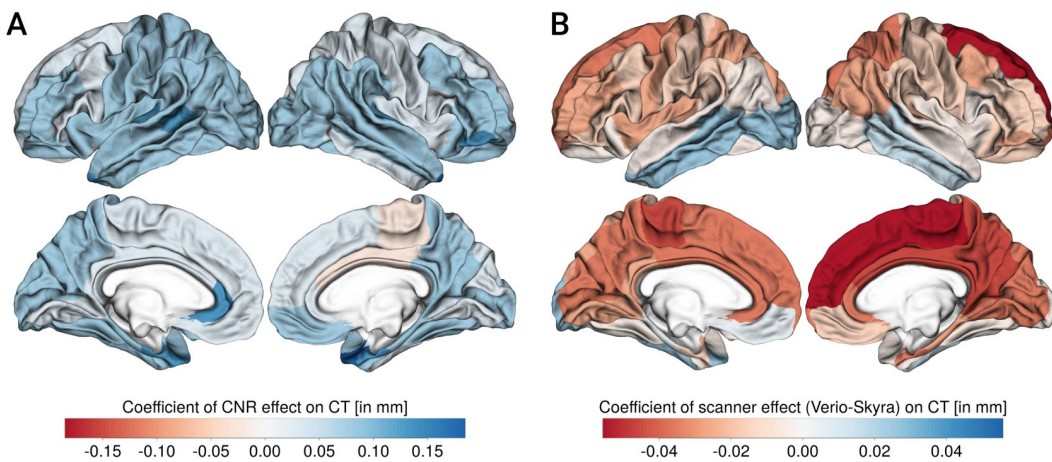

**Fig 9.** Association of CNR (A) and scanner (B, negative values indicate Skyra>Verio) with cortical thickness (CT), quantified as coefficients of a linear mixed model including both terms. Left column shows lateral and medial view of left hemisphere, right column shows lateral and medial view of right hemisphere.

CV we saw a gyrification dependent pattern, with higher CA and CV in sulci for Verio compared to Skyra, and higher CA and CV in gyri for Skyra compared to Verio. These differences shaped the observed ICC estimates. Gray-to-white matter CNR, but not SNR differed between the scanners, yet in contrast to previous studies, we found higher CNR on the (older) Verio scanner [37,38]. Still, while we found higher CNR to be associated with higher CT, the CT bias pattern was independent of differences in global measures of image quality [11,37].

Similarly, while gradient distortion correction reduced the bias and increased reliability for cortical measures, the overall medial-frontal to lateral-occipital bias in CT and the gyrification-dependent pattern in CA and CV remained similar. Thus, while gradient distortions impact the reliability of GM estimates, they do not fully explain the bias between Skyra and Verio scanner. Instead, we speculate that differences in scaling or signal intensities, which might cause altered white and gray matter contrast, have led to the observed differences [40]. This would be compatible with both the fronto-medial to occipital-lateral bias pattern (medial and subcortical regions biased toward higher CT and GM volume in Skyra compared to Verio) and the bias following the gyrification in CA. Upon visual inspections of the longitudinal runs (i.e. when both had been registered to a common template), we noticed a subtle expansion of the brain in Skyra compared to Verio in exemplary subjects. Taken together, we believe the systematic biases between Verio and Skyra stem from both scaling and image intensity differences, and are related to both differences in scanner hardware, e.g. receiver head coils, and software, e.g. differences in reconstruction algorithms.

While our results certainly overestimate the effects of a real upgrade as discussed above, they still support previous studies on the biasing effects of a scanner upgrade and urge for the use of an adequate correction method if an upgrade becomes necessary during a longitudinal study. One possibility is to measure the same subjects shortly before and after the upgrade and to derive scaling factors like in [16]. Another possibility, which does not require additional data acquisition, is longitudinal ComBat correction, which takes into account biased mean and scaling due to systematic scanner differences [41] or the use of a deep-learning-based harmonization framework [42].

## Limitations

The main limitation of our study is that we did not assess the impact of a true upgrade (i.e. repeated measurements on the same scanner), instead we performed a site-comparison in

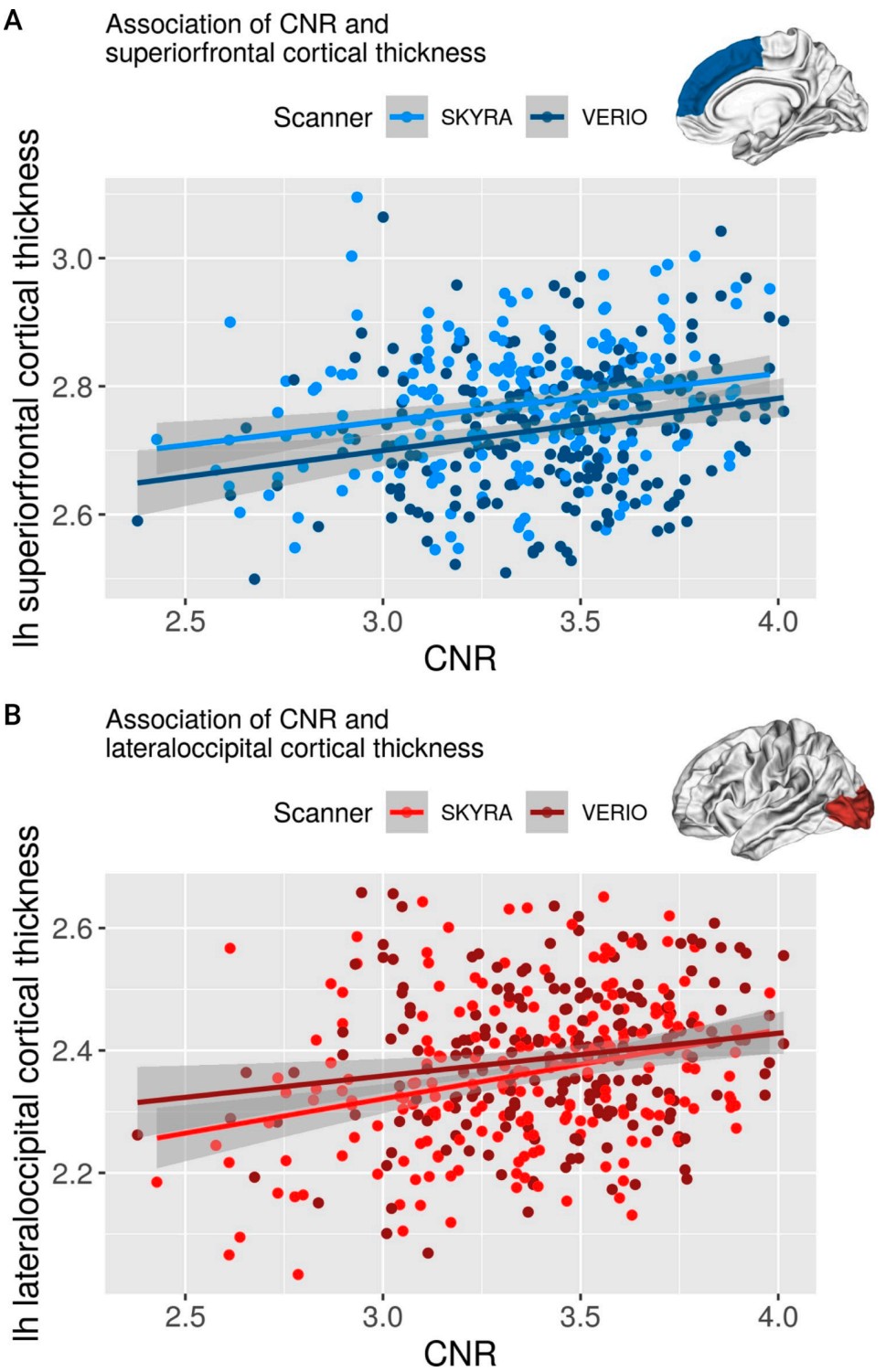

**Fig 10.** Association of CNR and cortical thickness in left superior frontal (A) and lateral occipital cortex (B).

which the MRI scanners at the two sides were as similar as possible. Another limitation is that we did not randomize the order of participants across scanners and that we could not assess

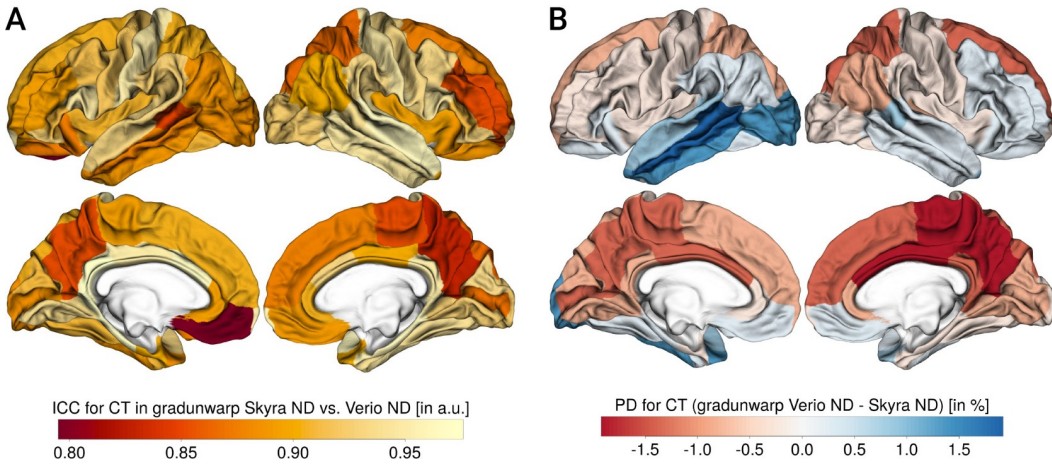

**Fig 11. Comparison of cortical thickness (CT) results from gradunwarp-corrected data.** Panel A: CT ICC, Panel B: CT PD (for each panel, left column shows lateral and medial view of left hemisphere, right column shows lateral and medial view of right hemisphere), negative values: Skyra>Verio, positive values: Verio>Skyra.

test-retest reliability as we only performed one scan on each system. Yet, previous studies indicated that PD of cortical and subcortical GM are comparable to our results (i.e. PD for subcortical volumes around 2–4% on Skyra and Verio scanners [17,43]). Finally, ICC is a common yet somewhat flawed measure of reliability. ICC does not reflect differences in inter-individual variability, as underlined by Bland-Altman plots of subcortical volumes, and was high in this study even though substantial bias was present.

## Strengths

Our study includes around 10 times more participants than previous reliability studies [37,38]. This gave us the power to detect small-to-medium systematic differences. For example,

**Table 3. Reliability (mean ICC, lower and upper ICC 95% confidence interval) and percent difference (PD) for subcortical grey matter volumes from gradient nonlinearity corrected data, separated by hemisphere (T<0 reflects Skyra>Verio, T>0 reflects Verio>Skyra).**

| Region of interest | hemi | ICC | lower ICC | upper ICC | PD | T | p | FDR-corrected p |
|---|---|---|---|---|---|---|---|---|
| Thalamus | Left | 0.97 | 0.96 | 0.97 | 1.95 | -13.50 | 0.00 | **0.00** |
| Thalamus | Right | 0.98 | 0.96 | 0.98 | 1.66 | -10.84 | 0.00 | **0.00** |
| Caudate | Left | 0.99 | 0.99 | 0.99 | 1.44 | -8.32 | 0.00 | **0.00** |
| Caudate | Right | 0.99 | 0.99 | 0.99 | 1.79 | -13.26 | 0.00 | **0.00** |
| Putamen | Left | 0.98 | 0.98 | 0.98 | 1.61 | -7.06 | 0.00 | **0.00** |
| Putamen | Right | 0.99 | 0.98 | 0.99 | 1.39 | -8.51 | 0.00 | **0.00** |
| Pallidum | Left | 0.97 | 0.97 | 0.98 | 2.50 | -2.84 | 0.01 | 0.01 |
| Pallidum | Right | 0.94 | 0.93 | 0.96 | 2.89 | -2.29 | 0.02 | 0.02 |
| Hippocampus | Left | 0.95 | 0.95 | 0.97 | 2.28 | -13.79 | 0.00 | **0.00** |
| Hippocampus | Right | 0.96 | 0.95 | 0.97 | 1.91 | -10.45 | 0.00 | **0.00** |
| Amygdala | Left | 0.91 | 0.89 | 0.92 | 3.58 | -2.93 | 0.00 | **0.00** |
| Amygdala | Right | 0.94 | 0.92 | 0.94 | 3.04 | -2.91 | 0.00 | **0.00** |
| Accumbens | Left | 0.81 | 0.78 | 0.85 | 9.23 | -8.84 | 0.00 | **0.00** |
| Accumbens | Right | 0.94 | 0.92 | 0.95 | 4.12 | -3.75 | 0.00 | **0.00** |

P-values are shown uncorrected, and FDR-corrected where **bold** indicates p<0.05.

Cohen's d of superior-frontal CT difference was -0.33, which requires a minimum number of 73 subject pairs to detect this effect with 80% power at $p = 0.05$. In population neuroimaging studies such as LIFE-Adult, we are interested in small effects, which is why it is relevant to assess systematic bias in an adequately powered sample. Another strength of our study is that we applied region-and brain-wide analyses, adjusted for gradient distortions and calculated complementary measures of reliability. Additionally, we present quantitative quality control measures derived from mriqc, a state-of-the-art quality control software.

## Conclusions

Taken together, in this study, we investigated the impact of a scanner upgrade on longitudinal cortical and subcortical GM measures. We found high reliability but strong regional biases in most regions of interest. While we possibly overestimated the effects of a real upgrade, this study urges for careful monitoring of scanner upgrades and adjustment of biases in longitudinal imaging studies. This may be achieved by deriving scaling factors immediately before/after the upgrade or by using longitudinal batch correction.

## Supporting information

**S1 File.**
(PDF)

## Author Contributions

**Conceptualization:** Evelyn Medawar, Ronja Thieleking, Arno Villringer, A. Veronica Witte, Frauke Beyer.

**Data curation:** Evelyn Medawar, Ronja Thieleking, Iryna Manuilova.

**Formal analysis:** Evelyn Medawar, Frauke Beyer.

**Funding acquisition:** Maria Paerisch, Arno Villringer, A. Veronica Witte.

**Investigation:** Evelyn Medawar, Ronja Thieleking, Maria Paerisch, A. Veronica Witte.

**Methodology:** Frauke Beyer.

**Project administration:** Ronja Thieleking, Iryna Manuilova, Maria Paerisch.

**Resources:** Arno Villringer, A. Veronica Witte.

**Software:** Frauke Beyer.

**Supervision:** A. Veronica Witte, Frauke Beyer.

**Validation:** Frauke Beyer.

**Visualization:** Evelyn Medawar, A. Veronica Witte, Frauke Beyer.

**Writing – original draft:** Frauke Beyer.

**Writing – review & editing:** Evelyn Medawar, A. Veronica Witte.

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
