## [Decision Letter · Decision Letter 0]

1 Apr 2021

PONE-D-20-28354

Estimating the effect of a scanner upgrade on measures of grey matter structure for longitudinal designs

PLOS ONE

Dear Dr. Beyer,

Thank you for submitting your manuscript to PLOS ONE. After careful consideration, we feel that it has merit but does not fully meet PLOS ONE’s publication criteria as it currently stands. Therefore, we invite you to submit a revised version of the manuscript that addresses the points raised during the review process.

Sorry about the long handling time of the review process. Both reviewers agreed that the scope of the work was very limited, but nevertheless thought it was interesting. The main limitation being that only two systems were compared. The reviewers had several comments and suggestion on how the manuscript can be improved. Please read and adjust carefully as required.

We look forward to receiving your revised manuscript.

Kind regards,

Peter Lundberg

Academic Editor

PLOS ONE

Journal Requirements:

1. Please ensure that your manuscript meets PLOS ONE's style requirements, including those for file naming. The PLOS ONE style templates can be found athttps://journals.plos.org/plosone/s/file?id=wjVg/PLOSOne_formatting_sample_main_body.pdf and https://journals.plos.org/plosone/s/file?id=ba62/PLOSOne_formatting_sample_title_authors_affiliations.pdf

Additional Editor Comments (if provided):

Reviewers' comments:

Reviewer's Responses to Questions

**Comments to the Author**

1. Is the manuscript technically sound, and do the data support the conclusions?

Reviewer #1: Partly

Reviewer #2: Yes

2. Has the statistical analysis been performed appropriately and rigorously? 

Reviewer #1: No

Reviewer #2: Yes

3. Have the authors made all data underlying the findings in their manuscript fully available?

Reviewer #1: Yes

Reviewer #2: No

4. Is the manuscript presented in an intelligible fashion and written in standard English?

Reviewer #1: Yes

Reviewer #2: Yes

5. Review Comments to the Author

Reviewer #1: The authors have investigated the between scanner difference between a 3 T Skyra fit and a Verio for subcortical estimates using FreeSurfer longitudinal stream. 121 subjects were included and 121 were scanned at the verio and 116 of them were scanned at the Skyra fit. The authors find systematic larger gray matter volume on estimated from the Skyra data compared to the Verio system.

This is a well structured and nicely written report with an important aim to show the effect of scanner upgrade and the number of subject make it powerful for estimating the effect between the two scanners used in this papper. However, since it is limited to only have compared two specific MR scanners the scope of the paper becomes very limited and can only point on difference between the specific systems with the specific software versions.

I think the paper would gain impact if the authors focused more on the specific differences between the systems (i.e. effect of distortion correction. Other potential differences coils/trasmitt power etc). I have a numer of more specific questions that should be addressed. Pleas see below.

Major

System differences

Pleas add a description of what was the difference in the systems, e.g. were the same coil used for receive? What are the difference in transmit amplitud and changes, receive were the same coil used, number of channels.

The lac of information of what differs between the systems makes it difficult to evaluate what could cause the differences in gray matter volumes.

Comparing volumes

I have doubts about using ICC for comparing the estimated gray matter volumes the estimated ICC values seems to be strongly dependent on the range of the subjects volumes. I.e. the higher ICC for caudate compared to the ICC for amygdala could be that the subjects have a higher variation in caudate volume than the variation in amygdala volume. As I see it Bland Altman analyses would be more suited for comparing volumes estimate using the different systems. Pleas add and discuss Bland Altman plots for the gray matter volumes.

QA measures

It is interesting that the Verio scanner seems to perform better for all images quality than the Skyra fit scanner. However, I lac a discussion of what the underlying reason for this. It is also related to the lac of information of the systems. E.g. could this be due to differences that the MPRAGE sequence are optimised for the Verio system eg flip angle optimised for the transmit system of the Verio, grappa factor optimised for the verio head coil etc.

Figure 6,

I do not get the CNR data to match. In the left panel Skyra ND data goes between 3 to 3.4 and in the right panel the Skyra ND CNR values goes between 3.1 and 3.6.

Minor

Pleas present Age in Median and range, mean and sd are more suitable parameters that are normal distributed. The same goes for time between scanning.

Page 11 section Comparison to previous upgrade studies, In this section the gray matter volume are abbreviated some time as GM volume some times as GV some times as GMV please be consistent.

Reviewer #2: PONE-D-20-28354 / reviewer comments

Thank you for giving me the opportunity to review your manuscript with the title “Estimating the effect of a scanner upgrade on measures of grey matter structure for longitudinal designs”.

This manuscript investigates the potential impact of magnetic resonance scanner upgrades during longitudinal studies, here specifically on the impact of grey matter structure.

The changes were observed on a rather large cohort of 116 participants being scanned on two different MR scanners from the same vendor field strength.

It was found the scan-recan reliability was good as well as the scan-recan difference but differences were observed in certain cortical and subcortical regions.

As a conclusion, the authors recommend to account for these effects when planning for a scanner upgrade.

The conclusion might not be novel but because of the large cohort used it gives this study the statistical power needed to draw conclusions. What makes this manuscript also original is that the older of the two scanners is a predecessor model of the newer scanner.

I however think that a better structuring of the Methods and Analysis section would be helpful for the reader where the performed analyses and metrics are clearer introduced. In particular, a “main analysis” is introduced but it is not mentioned in the Results section, but only in the discussion.

Major comments

1. The authors are analyzing the data here in two ways (cf. Fig 1.) by comparing Verio ND to Syra D data and by comparing Verio ND with gradunwarp to Skyra ND with gradunwarp. This is introduced on page 4, line 99-102. However, these lines are in the subsection on details regarding the performed MRI acquisitions.

a. I believe it makes much more sense that this is included in the Analysis section which allows then to list all the different analyses performed using the described metrics (ICC, PD, etc…).

b. The first comparison, referred to as “main analysis” in the manuscript (Imaging sequence subsection) is however done on data from the Skyra with gradient distortions correction and from the Verio without gradient distortion correction. As pointed out by the authors themselves, cf. subsection “Gradient distortion correction”, gradient distortion correction has an impact on the final image, as described in referenced literature, so I am missing the reason to do this comparison here.

I would have found more logical to compare the data sets Skyra ND with Verio ND, because this makes a direct comparison between data which have seen supposedly the same steps during image reconstruction and make a nice introduction/motivation to the analysis described in subsection “Effect of offline gradient distortion correction”

I am therefore asking the authors to add this comparison and to motivate the reason to do a comparison between the data sets Skyra D with Verio ND.

2. Was the same coil used on both scanners? Please comment, discuss the possible implications with your findings. I speculate this could have a strong influence on the differences observed regionally and why you do not have the same findings as in the other studies referenced in the discussion, [33] and [34].

3. Was the same coil combination algorithm used (sum of squares or adaptive coil combination) which is available on both scanners?

I am aware that the coil combination algorithm has evolved from VB to VE which should not impact so much MPRAGE images in the brain, but is an example that not only hardware changes can have an effect but also pure software updates. In the discussion section, line 380, you are limiting yourself there to hardware. Purely software upgrades can also have impacts on the final obtained image from the scanner because of changes in the sequence or the image reconstruction.

4. As stated, it could have been adequate to add a test re-test on a subset of the participants on the two scanners to help to put perspective on the given numbers for the scan-rescan reliability. If such a data subset exists, it would really roundup this study nicely to address repeatability. Are there published articles which have investigated the repeatability of the MPRAGE on one of both scanners used in this study and where a similar analysis is performed on grey matter structure?

Minor comments

1. page 3 line 55: Siemens Healthcare MAGNETOM Verio to Skyrafit. Please consisting of the naming of the scanners throughout the manuscript

2. page 3 line 64: 3 T MAGNETOM Verio vs Skyrafit

3. page 3 line 67: Tim 4G, new bdy coil

4. page 3 line 66-67: the information in parentheses is not needed here or as an alternative the reference the product brochure which lists the hardware change https://cdn0.scrvt.com/39b415fb07de4d9656c7b516d8e2d907/1800000000279602/0c4ab340a772/MRI-MAGNETOM-Verio-Upgradebrochure_1800000000279602.pdf

5. page 3 line 68: on the upgrade of the Verio to a Skyrafit ([...]

6. page 3 line 68 to 70: Sentence is confusing here. Could be removed or integrated to the Discussion

7. page 4, line 80-81: the software versions VB and VE are not subsequent with each other.

8. page 4, line 81: Siemens Healthcare, Erlangen, Germany

9. page 4, line 95: remove reference to a paper not yet submitted, available

Repalce with e.g. " [...] analysis results will be presented in subsequent publication(s)."

10. page 4, line 110:"correction" should be removed, should'nt it?

11. line 120: add a reference to M.F. Glasser et al. / NeuroImage 80 (2013) 105–124 for the graduwarp code and J. Jovicich et al. / NeuroImage 30 (2006) 436–443 as a reference for the method itself

Is a version number of the used github code used needed?

12. line 130: remove replace ',' by 'and'

13. line 134: remove ','

14. line 146: in parenthenses (version 0.15 ...)

15. line 153: Shift "As outcomes" to the end of the sentence.

16. I found this paper

N.K. Dinsdale, M. Jenkinson and A.I.L. Namburete, NeuroImage 228 (2021)

which seems as well promising and should be cited alongside the ComBat method referenced in the discussion to tackle the problem described in your study.

17. It would be helpful that the abbreviations defined in the manuscript are written out again in the figure captions to help the reader to understand what e.g. ICC or PD stands for.

6. PLOS authors have the option to publish the peer review history of their article (what does this mean?). If published, this will include your full peer review and any attached files.

Reviewer #1: **Yes: **Anders Tisell

Reviewer #2: No

---

## [Author Response · Author response to Decision Letter 0]

20 May 2021

Answers to Reviewers

We thank both reviewers for their thorough review of our work and the very helpful recommendations. Please see below for our replies, printed in italics. Changes to the manuscript are highlighted in yellow. We changed the manuscript according to the reviewer’s suggestions and provide a version, in which all track changes are highlighted in yellow. 

Reviewer 1: 

1. System differences

Please add a description of what was the difference in the systems, e.g. were the same coil used for receive? What are the difference in transmit amplitud and changes, receive were the same coil used, number of channels.

The lac of information of what differs between the systems makes it difficult to evaluate what could cause the differences in gray matter volumes.

We agree that more information on the systems would make it easier for the reader to understand the differences we describe. We thus added a table which shows scanner and image acquisition parameters to the Methods section of the manuscript (see Table 1 below). Overall, the scanners differed in various technical parameters: software version, transmit coil and transmit-receive electronics, which is reflected in the transmission amplifier reference amplitude. Gradients and field strength were identically constructed, but might still slightly vary between scanners. Similarly, we used two 32-channel receive coils which differed in plug system and could thus not be interchanged. Sequence parameters were the same on both scanners, yet again, we do not have access to possible changes in reconstruction algorithms between software versions.

(here Table 1 should appear)

2. Comparing volumes

I have doubts about using ICC for comparing the estimated gray matter volumes the estimated ICC values seems to be strongly dependent on the range of the subjects volumes. I.e. the higher ICC for caudate compared to the ICC for amygdala could be that the subjects have a higher variation in caudate volume than the variation in amygdala volume. As I see it Bland Altman analyses would be more suited for comparing volumes estimate using the different systems. Pleas add and discuss Bland Altman plots for the gray matter volumes.

We agree with the reviewer that Bland Altman analyses are a suitable addition to our ICC analysis. We used ICC because it is a standard measure to assess agreement between raters, which has been used in other studies comparing GMV (test-retest) reliability [Plitman et al. 2020 and Potvin et al., 2019]. Yet, we are aware that ICC depends on the amount of between-person variance and may not be comparable when measures (e.g. amygdala and caudate volume) differ in variability across participants. Thus, we added exemplary Bland-Altman plots for superior frontal and lateral occipital cortical thickness (see Figures below), as well as amygdala and caudate to the main manuscript and a plot showing mean and difference in all subcortical volumes to the supplementary Material. In the results section for the cortical analysis on p. 16, it now reads: “Bland-Altman plots confirmed the bias of Verio versus Skyra measurements. Exemplary plots of superior frontal and lateral occipital regions are shown in Figure 6. For the superior frontal region, CT, CA and CV are larger for Skyra compared to Verio with 95% of CT differences were between -0.04 and -0.12 mm, 95% of CA differences were between -103.12 and -285.25 , and 95% of CV differences were between -694.23 and -1841.61 . The inverse pattern was present in the lateral occipital region with 95% of CT differences were between 0.03 and -0.07 , 95% of CA differences were between -55.32 and -156.55 , and 95% of CV differences were between -86.04 and -772.28 ”

On p. 17, we further added “Bland-Altman plots for subcortical regions confirmed the systematic bias and further indicated that differences in variability between subjects influenced ICC estimates. For example, there was high between-subject variability in the Thalamus, so that, despite large differences between measurements, ICC was high. Similarly, differences between scanners were less pronounced in the Accumbens, yet, due to lower between-subject variability the ICC of this region was lower (see Figure 7, and Figure 1 in the Supplementary Material). For the Accumbens, 95% of differences between scanners were between -143.02 and 37.38 , for Thalamus 95% of differences were between -914.4 and 233.85 .”

In the discussion on p.24, we now discuss the limitations of the ICC: “ICC is a common yet somewhat flawed measure of reliability. ICC does not reflect differences in inter-individual variability, as underlined by Bland-Altman plots of subcortical volumes, and was high in this study even though substantial bias was present.”

3. QA measures

It is interesting that the Verio scanner seems to perform better for all images quality than the Skyra fit scanner. However, I lac a discussion of what the underlying reason for this. It is also related to the lac of information of the systems. E.g. could this be due to differences that the MPRAGE sequence are optimised for the Verio system eg flip angle optimised for the transmit system of the Verio, grappa factor optimised for the verio head coil etc.

We agree with the reviewer and extended our analysis to signal-to-noise ratio which is also provided by mriqc. Here, we saw no significant SNR difference between Verio and Skyra. Yet, we found higher grey-to-white matter contrast as indicated by the higher CNR and CJV in Verio compared to Skyra.

The MPRAGE sequence from the ADNI-3 protocol is recommended for both Verio and Skyra (http://adni.loni.usc.edu/methods/documents/mri-protocols/) and was performed with identical settings in this study. Thus, we do not believe that the quality differences relate to specific settings in the sequence such as flip angle or acceleration factor. Instead, the small, yet significant differences in gray-white matter contrast likely arise from variation in transmit and receive hardware as well as software. We added Table 1 to describe differences between scanners in more detail on p. 8. The analysis of SNR is described in the Methods section on p.11: “We used signal-to-noise ratio (SNR) to assess overall signal quality and compared contrast-to-noise ratio (CNR) to quantify the difference between grey and white matter intensity distributions. Furthermore, we used coefficient of joint variation (CJV) which also reflects grey-to-white matter contrasts and entropy focus criterion (EFC) to describe the amount of ghosting and blurring induced by head motion [24].” and in the Results section on p. 18: “We found that there was no significant difference in SNR between Verio and Skyra (=0.06, p = 0.07).”

4. Figure 6

I do not get the CNR data to match. In the left panel Skyra ND data goes between 3 to 3.4 and in the right panel the Skyra ND CNR values goes between 3.1 and 3.6.

You are right, thank you for pointing this out, there was an error in the generation of the plots. According to the general change in the structure of the manuscript (see Answer to Reviewer 2), we now present Skyra ND versus Verio ND as the main analysis, and moved the comparison of Skyra D and Verio ND to the supplementary Material (p.14). Now, Skyra D CNR values are the same in both panels.

Minor

5. Please present Age in Median and range, mean and sd are more suitable parameters that are normal distributed. The same goes for time between scanning.

We updated the presentation of age and time between scanning in the Methods section, please see p. 6, where it now reads: “121 healthy participants (median age = 28 years, range = 19 - 54 years [...]” and “The median time between sessions was 1.92 months (range: 0.12 - 4.56 months)” 

6. Page 11 section Comparison to previous upgrade studies, In this section the gray matter volume are abbreviated some time as GM volume some times as GV some times as GMV please be consistent.

Thank you for spotting this issue. We modified the manuscript to be more consistent.

Reviewer #2

Major comments

1. The authors are analyzing the data here in two ways (cf. Fig 1.) by comparing Verio ND to Syra D data and by comparing Verio ND with gradunwarp to Skyra ND with gradunwarp. This is introduced on page 4, line 99-102. However, these lines are in the subsection on details regarding the performed MRI acquisitions.

a. I believe it makes much more sense that this is included in the Analysis section which allows then to list all the different analyses performed using the described metrics (ICC, PD, etc…).

b. The first comparison, referred to as “main analysis” in the manuscript (Imaging sequence subsection) is however done on data from the Skyra with gradient distortions correction and from the Verio without gradient distortion correction. As pointed out by the authors themselves, cf. subsection “Gradient distortion correction”, gradient distortion correction has an impact on the final image, as described in referenced literature, so I am missing the reason to do this comparison here.

I would have found more logical to compare the data sets Skyra ND with Verio ND, because this makes a direct comparison between data which have seen supposedly the same steps during image reconstruction and make a nice introduction/motivation to the analysis described in subsection “Effect of offline gradient distortion correction”

I am therefore asking the authors to add this comparison and to motivate the reason to do a comparison between the data sets Skyra D with Verio ND.

We agree with the points the reviewer raised. We moved the description of the performed comparisons to the section “Analysis”, and changed the main analysis to be the comparison between Verio ND and Skyra ND. We also believe that it makes more sense to compare images at a similar stage of image reconstruction, especially knowing that gradient distortions can have a large impact on image quality. On p. 19, this is illustrated by the comparison of ICC between Verio ND vs Skyra ND, gradunwarp Verio ND vs. Skyra ND and the “default” analysis Verio ND vs Skyra D. “In addition, we saw that the systematic bias was largest when comparing the default scanner output images Skyra D (online gradient distortion correction) and Verio ND (mean ICC gradunwarp Skyra ND vs Verio ND: 0.91, mean ICC Skyra ND vs Verio ND: 0.91, mean ICC Skyra D vs Verio ND: 0.89, ANOVA F-test: F = 3.7, p = 0.026).” Here, Verio ND vs. Skyra D performed slightly, yet significantly worse. 

Please find the modified Figure 1 on p. 11 and below, and all changes in the updated manuscript highlighted in yellow.

2. Was the same coil used on both scanners? Please comment, discuss the possible implications with your findings. I speculate this could have a strong influence on the differences observed regionally and why you do not have the same findings as in the other studies referenced in the discussion, [33] and [34].

Yes, a 32-channel head coil was used on both scanners. We could not use the same head coil on both, because the scanners differed in plug system. We thus added the possibility that head coils influenced our results to the Discussion on p. 23, where it now reads: “Taken together, we believe the systematic biases between Verio and Skyra stem from both scaling and image intensity differences, and are related to both differences in scanner hardware, e.g. receiver head coils, and software, e.g. differences in reconstruction algorithms.”

3. Was the same coil combination algorithm used (sum of squares or adaptive coil combination) which is available on both scanners?

I am aware that the coil combination algorithm has evolved from VB to VE which should not impact so much MPRAGE images in the brain, but is an example that not only hardware changes can have an effect but also pure software updates. In the discussion section, line 380, you are limiting yourself there to hardware. Purely software upgrades can also have impacts on the final obtained image from the scanner because of changes in the sequence or the image reconstruction.

We added Table 1 to the manuscript to give the reader an overview of differences between scanners and acquisitions (see also reply to comment 1 of reviewer #1). The same coil combination algorithm (adaptive coil combination) was used for both. We however agree that software upgrades are important and extended the discussion accordingly. It now reads, for example on p. 23:”Taken together, we believe the systematic biases between Verio and Skyra stem from both scaling and image intensity differences, and are related to both differences in scanner hardware, e.g. receiver head coils, and software, e.g. differences in reconstruction algorithms.”

4. As stated, it could have been adequate to add a test re-test on a subset of the participants on the two scanners to help to put perspective on the given numbers for the scan-rescan reliability. If such a data subset exists, it would really roundup this study nicely to address repeatability. Are there published articles which have investigated the repeatability of the MPRAGE on one of both scanners used in this study and where a similar analysis is performed on grey matter structure?

We agree with the reviewer that a retest-acquisition of the same participants on the same scanner would have been ideal to round up this study and assess test-retest reliability. Unfortunately, we did not have the opportunity to acquire this data due to time and personnel constraints, i.e. that the follow-up of the cohort study was scheduled already. Yet, previous studies have reported percent difference values for cortical thickness from 1-5% and for subcortical volumes from 2 – 4 % (but no bias) for test-retest acquisitions on Skyra and Verio (Jovivich et al., 2013 and Yan et al., 2020). 

We have expanded the Limitations section in the manuscript (p. 24) accordingly, where it now reads: “Another limitation is that we did not randomize the order of participants across scanners and that we could not assess test-retest reliability as we only performed one scan on each system. Yet, previous studies indicated that PD of cortical and subcortical GM are comparable to our results (i.e. PD for subcortical volumes around 2-4 % on Skyra and Verio scanners.“

Minor comments

We adressed all minor comments by the reviewer throughout the manuscript.

1. page 3 line 55: Siemens Healthcare MAGNETOM Verio to Skyrafit. Please consisting of the naming of the scanners throughout the manuscript.

We changed to Verio and Skyra whenever referring to the scanners. We give the exact product names in the Methods section and added “Scanners are referred to as Verio and Skyra throughout the manuscript.”.

2. page 3 line 64: 3 T MAGNETOM Verio vs Skyrafit

3. page 3 line 67: Tim 4G, new bdy coil

4. page 3 line 66-67: the information in parentheses is not needed here or as an alternative the reference the product brochure which lists the hardware change https://cdn0.scrvt.com/39b415fb07de4d9656c7b516d8e2d907/1800000000279602/0c4ab340a772/MRI-MAGNETOM-Verio-Upgradebrochure_1800000000279602.pdf

We included more details on hardware and software changes in Table 1 as recommended by both reviewers, and referenced the brochure stored in our github repository https://github.com/fBeyer89/life_upgrade.

5. page 3 line 68: on the upgrade of the Verio to a Skyrafit ([...]

6. page 3 line 68 to 70: Sentence is confusing here. Could be removed or integrated to the Discussion. The sentence has been removed.

7. page 4, line 80-81: the software versions VB and VE are not subsequent with each other. We removed this part of the sentence.

8. page 4, line 81: Siemens Healthcare, Erlangen, Germany

9. page 4, line 95: remove reference to a paper not yet submitted, available

Repalce with e.g. " [...] analysis results will be presented in subsequent publication(s)."

10. page 4, line 110:"correction" should be removed, should'nt it? No, it’s the correction of gradient distortions.

11. line 120: add a reference to M.F. Glasser et al. / NeuroImage 80 (2013) 105–124 for the graduwarp code and J. Jovicich et al. / NeuroImage 30 (2006) 436–443 as a reference for the method itself. We added the references to the manuscript.

12. line 130: remove replace ',' by 'and'

13. line 134: remove ','

14. line 146: in parenthenses (version 0.15 ...)

15. line 153: Shift "As outcomes" to the end of the sentence.

16. I found this paper

N.K. Dinsdale, M. Jenkinson and A.I.L. Namburete, NeuroImage 228 (2021)

which seems as well promising and should be cited alongside the ComBat method referenced in the discussion to tackle the problem described in your study. We agree, and included the paper in the Discussion on p. 16.

17. It would be helpful that the abbreviations defined in the manuscript are written out again in the figure captions to help the reader to understand what e.g. ICC or PD stands for. We adapted the figure captions accordingly.

---

## [Decision Letter · Decision Letter 1]

7 Jul 2021

PONE-D-20-28354R1

Estimating the effect of a scanner upgrade on measures of grey matter structure for longitudinal designs

PLOS ONE

Dear Dr. Beyer,

Thank you for submitting your manuscript to PLOS ONE. After careful consideration, we feel that it has merit but does not fully meet PLOS ONE’s publication criteria as it currently stands. Therefore, we invite you to submit a revised version of the manuscript that addresses the points raised during the review process.

There are a few remaining issues, they are of minor nature but nevertheless need to be corrected. Please see attached comments from both reviewers.

We look forward to receiving your revised manuscript.

Kind regards,

Peter Lundberg

Academic Editor

PLOS ONE

Journal Requirements:

Reviewers' comments:

Reviewer's Responses to Questions

**Comments to the Author**

1. If the authors have adequately addressed your comments raised in a previous round of review and you feel that this manuscript is now acceptable for publication, you may indicate that here to bypass the “Comments to the Author” section, enter your conflict of interest statement in the “Confidential to Editor” section, and submit your "Accept" recommendation.

Reviewer #1: All comments have been addressed

Reviewer #2: All comments have been addressed

2. Is the manuscript technically sound, and do the data support the conclusions?

Reviewer #1: Yes

Reviewer #2: Yes

3. Has the statistical analysis been performed appropriately and rigorously? 

Reviewer #1: Yes

Reviewer #2: Yes

4. Have the authors made all data underlying the findings in their manuscript fully available?

Reviewer #1: Yes

Reviewer #2: Yes

5. Is the manuscript presented in an intelligible fashion and written in standard English?

Reviewer #1: Yes

Reviewer #2: Yes

6. Review Comments to the Author

Reviewer #1: The authors have addressed all my concerns, and this version is more readable and better structured. I only have minor comments regarding the Bland Altman plots in figure 6 and 7.

There are a bit much text in the figure that make it hard to read. E.g. The region name could be moved from being repeated 3 times in figure 6 the title of the x-axis to be written one time above each column. Likewise the title of the y-axis could be omitted for the second column (both figure 6 and 7).

Moreover, the figure would bee more readable if the scale of the y-axis were the same for all regions and the bias would be more readable if the scales were centred around 0 figure (both figure 6 and 7).

In figure 6 the y-label is cooped and it is started that the distance is in m not mm.

Reviewer #2: Dear author, thank you for addressing the reviewers’ comments. I have only some minor comments on the revised manuscript which you will find below.

Thank you for the interesting manuscript.

Minor comments:

Table 1, System software version : Please add, if possible, the complete software version name, e.g. syngo MR B17A or E11A. For example, there is a big difference between the versions E11C and E11A. Please update correspondingly line 81 and 82. Given the dates, I believe that it must be B17A and E11A.

Table 1, Tx/Rx coil line: Please state the correct number of RF channel for your systems. You are listing the various RF configuration possibilities. It is for the Verio either [102x8], [102x18] or [102x32]. Because you are using the 32 channel head coil it can only the [102x32] RF configuration. Same for the Skyra, it can only be one of the two listed RF configuration.

Line 55: from MAGNETOM Verio to MAGNETOM Skya fit.

Line 81 and 82: Magnetom in capital letters.

Line 81 and 82: please change to “ […] MAGNETOM Verio with software version syngo MR B17 ([…] “ and “ […] MAGNETOM Verio with software version syngo MR E11 ([…] “. Please add the correct software version as in Table 1.

Line 85 to 86: would weeks not be a better unit instead of months?

Line 96: “Further, the scanner were not issue of an actual […]”

Could it be that part of the sentence is missing?

Line 134: Here you write gradunwrap not in italic but in line 140 you write it in italic. In order to be consistent throughout the manuscript, I suggest introducing the names of all used software package in italic, e.g. FreeSurfer, psy, etc…

Line 276: Two points after each other, only one is needed to finish the sentence.

Line 401: no comma after “one study”. Please cite the paper of this study separately for clarity.

7. PLOS authors have the option to publish the peer review history of their article (what does this mean?). If published, this will include your full peer review and any attached files.

Reviewer #1: **Yes: **Anders Tisell

Reviewer #2: No

---

## [Author Response · Author response to Decision Letter 1]

19 Jul 2021

Answers to Reviewers

We thank the reviewers for their positive evaluation and the additional comments to the revised version. We changed the manuscript according to the reviewer’s suggestions and provide a version, in which all track changes are highlighted in yellow. 

Reviewer #1: The authors have addressed all my concerns, and this version is more readable and better structured. I only have minor comments regarding the Bland Altman plots in figure 6 and 7.

There are a bit much text in the figure that make it hard to read. E.g. The region name could be moved from being repeated 3 times in figure 6 the title of the x-axis to be written one time above each column. Likewise the title of the y-axis could be omitted for the second column (both figure 6 and 7).

Moreover, the figure would bee more readable if the scale of the y-axis were the same for all regions and the bias would be more readable if the scales were centred around 0 figure (both figure 6 and 7).

In figure 6 the y-label is cooped and it is started that the distance is in m not mm.

We thank the reviewer for the suggestions which indeed made the figures more easily readable. We abstained from making the y-axis the same for all regions, because the scales of CT, CA and CV differ largely. Yet, we used the same y-axis limits for both cortical regions, and all subcortical regions. Please see the updated figures in the manuscript.

Reviewer #2: Dear author, thank you for addressing the reviewers’ comments. I have only some minor comments on the revised manuscript which you will find below. Thank you for the interesting manuscript.

Minor comments: Table 1, System software version : Please add, if possible, the complete software version name, e.g. syngo MR B17A or E11A. For example, there is a big difference between the versions E11C and E11A. Please update correspondingly line 81 and 82. Given the dates, I believe that it must be B17A and E11A.

Thank you for pointing out the importance of the exact version specification. We updated the version names to syngo MR B17A and E11C throughout the manuscript. 

Table 1, Tx/Rx coil line: Please state the correct number of RF channel for your systems. You are listing the various RF configuration possibilities. It is for the Verio either [102x8], [102x18] or [102x32]. Because you are using the 32 channel head coil it can only the [102x32] RF configuration. Same for the Skyra, it can only be one of the two listed RF configuration.

We apologize for this mistake. We changed the table to include the correct RF configurations [102 x 32] for Verio and [204x64] for Skyra.

Line 55: from MAGNETOM Verio to MAGNETOM Skya fit.

Line 81 and 82: Magnetom in capital letters.

Line 81 and 82: please change to “ […] MAGNETOM Verio with software version syngo MR B17 ([…] “ and “ […] MAGNETOM Verio with software version syngo MR E11 ([…] “. Please add the correct software version as in Table 1.

Line 85 to 86: would weeks not be a better unit instead of months?

We changed the manuscript according to the reviewer’s suggestions.

Line 96: “Further, the scanner were not issue of an actual […]” Could it be that part of the sentence is missing?

We clarified this sentence by changing it to “In this study, the Verio scanner did not undergo an actual upgrade and the scanners therefore differed in the main B0-field and other hardware components.”

Line 134: Here you write gradunwrap not in italic but in line 140 you write it in italic. In order to be consistent throughout the manuscript, I suggest introducing the names of all used software package in italic, e.g. FreeSurfer, psy, etc… Line 276: Two points after each other, only one is needed to finish the sentence.

We agree with the reviewer and changed all software tools and packages to be shown in italics.

Line 401: no comma after “one study”. Please cite the paper of this study separately for clarity.

We removed the comma and cited the studies separately as suggested.

---

## [Editor Report · Decision Letter 2]

16 Aug 2021

Estimating the effect of a scanner upgrade on measures of grey matter structure for longitudinal designs

PONE-D-20-28354R2

Dear Dr. Beyer,

We’re pleased to inform you that your manuscript has been judged scientifically suitable for publication and will be formally accepted for publication once it meets all outstanding technical requirements.

Kind regards,

Peter Lundberg

Academic Editor

PLOS ONE
---

## [Editor Report · Acceptance letter]

24 Sep 2021

PONE-D-20-28354R2 

Estimating the effect of a scanner upgrade on measures of grey matter structure for longitudinal designs 

Dear Dr. Beyer:

I'm pleased to inform you that your manuscript has been deemed suitable for publication in PLOS ONE. Congratulations! Your manuscript is now with our production department. 

Kind regards, 

on behalf of

Professor Peter Lundberg 

Academic Editor

PLOS ONE